# Microbiome of the Aphid Genus *Dysaphis* Börner (Hemiptera: Aphidinae) and Its Relation to Ant Attendance

**DOI:** 10.3390/insects13121089

**Published:** 2022-11-26

**Authors:** Natalia Kaszyca-Taszakowska, Łukasz Depa

**Affiliations:** Institute of Biology, Biotechnology and Environmental Protection, Faculty of Natural Sciences, University of Silesia in Katowice, 40-007 Katowice, Poland

**Keywords:** symbiont, mutualism, bacterial communities, ecology, evolution, host plants

## Abstract

**Simple Summary:**

Many organisms are supported by cooperative bacteria, the so-called microbiome, which enhance their survival abilities. Aphids also have many bacterial symbionts, providing them with nutritional components as well as protection against environmental conditions and pathogens or natural enemies. Similar relation connects aphids with ants, so the mutual reliance of bacteria and ants may be exclusive to aphids. Also, the relationship of aphids with their host plants may be decisive in shaping the aphid microbiome. Here we showed, that aphids attended by ants have less but more diverse bacterial symbionts while aphids on their primary host plant, where they reproduce sexually, have fewer symbionts than on secondary host plants. The results suggest that symbiosis with ants may serve aphids to replace some functions of bacterial symbionts. However, it seems that the relationship between aphids and host plants is a much stronger trait influencing the composition of the aphid microbiome.

**Abstract:**

Among mutualistic relationships of aphids with other organisms, there are two that seem to be of major importance: trophobiosis with ants and endosymbiosis of bacteria. While the former is well studied, the latter is the subject of an increasing amount of research constantly revealing new aspects of this symbiosis. Here, we studied the possible influence of ant attendance on the composition of aphid microbiota on primary and secondary hosts exploited by the aphid genus *Dysaphis*. The microbiome of 44 samples representing 12 aphid species was studied using an Illumina HiSeq 4000 with the V3-V4 region of 16S rRNA. The results showed a higher abundance of common facultative symbionts (*Serratia*, *Regiella*, *Fukatsuia*) in aphid species unattended by ants, but also on secondary hosts. However, in colonies attended by ants, the general species composition of bacterial symbionts was more rich in genera than in unattended colonies (*Wolbachia*, *Gilliamella*, *Spiroplasma*, *Sphingomonas*, *Pelomonas*). The results indicate a huge variability of facultative symbionts without clear correlation with ant attendance or aphid species. The possibility of multiple routes of bacterial infection mediated by ant-made environmental conditions is discussed.

## 1. Introduction

Animals exhibit diverse relationships with different types of symbiotic bacteria which are common in insects and can greatly influence their ecology and evolution [1]. In the case of aphids, such an obligate bacterial symbiont is *Buchnera aphidicola*, which established its symbiotic relation with aphids more than 100 Mya and undergoes constant genome reductions [2,3]. This symbiont provides aphids with amino acids and is transmitted maternally, coevolving with particular aphid lineages [4,5]. In contrast to the obligate symbionts, the facultative bacterial partners only deliver fitness benefits in the context of specific ecological conditions. In aphids, there are several facultative endosymbionts that are considered to be relatively common as well as serving as various ecological adaptations [6], e.g., *Serratia*, *Hamiltonella*, *Wolbachia*, *Regiella*, and *Fukatsuia*. The functions of particular secondary endosymbionts vary and may include an increase in resistance to heat, parasitoids, and fungal infections, enhancing host plant fitness and fecundity as well as decreasing longevity, growth, reproduction, defensive behaviours, or host plant fitness [7].

Therefore, being facultative secondary symbionts, these bacteria seem to be acquainted and have been lost several times repeatedly in evolutionary history of aphids, e.g., genus *Cinara* [8]. This may be proven also by the geographical differentiation of the endosymbiotic biome of aphids [6] and also a clear variety of bacterial taxa involved in this symbiotic relationship [9,10,11]. Moreover, horizontal transfer of facultative endosymbionts may play a role in the acquisition of adaptative capabilities by aphids [12]. However, research conducted thus far indicates a strong influence of host plant as well as aphid phylogeny on the composition of the bacterial microbiome of aphids. Closely related aphid species have more similar bacterial communities. This indicates strong co-adaptation between aphids and their facultative symbionts [13], but other, e.g., ecological factors and their influence on the aphid microbiome remain insufficiently studied. It has been proven that general plant species richness in habitats may influence the microbiome diversity in aphids [14,15], but the mechanism of this influence is not clear.

Apart from host plant richness, the presence of ants may be an additional factor influencing the composition of aphid microbiome. It has been reported that the presence of ants tending aphids may alter the composition of the aphid microbiome by changing the abundance of particular symbionts. Studies by Mandrioli et al. [16] showed that the presence of ants may influence the abundance of *Hamiltonella*, with facultative myrmecophilic aphids having higher amount of *Hamiltonella* than obligatory ones. This is based on the balance of costs and benefits between defensive role of ants and bacteria, wherein keeping a high amount of bacteria in the case of a constant presence of protective ants is less cost effective for aphids. Moreover, Henry et al. [17] indicated that ant-tending decreases the proportion of *Hamiltonella* and *Regiella* but increases the proportion of *Serratia.* Although generally ant-tended aphids have less facultative symbionts, it may also be influenced by aphid relations with host plants. Moreover, the effect of ant bacteria on the microbiome of aphids may not be excluded [18].

The above-mentioned research focused either on many unrelated aphid taxa, belonging to various subfamilies [16,17], or on species with not fully recognised life cycles and polymorphisms (*Prociphilus*, [18,19]). The exact role of myrmecophily in shaping the aphid microbiome remains unknown. To shed some light on the possible influence of ants on the aphid microbiome, we researched the structure of bacterial endosymbionts in the aphid genus *Dysaphis* Börner. This palearctic aphid genus is very diverse, both taxonomically—with more than 100 species described—as well as ecologically. It comprises both heteroecious as well as monoecious species, monophagous as well as oligophagous, primary sexually reproducing but also permanently parthenogenetic, and mostly ant-tended but also unattended taxa [19]. Such differentiated adaptations pose some difficulties in proper species recognition within this genus. Due to the significant importance of *Dysaphis* in agriculture, with some of the species being serious pests of cultivated plants, it seemed to be quite important to study the composition of the microbiome of this genus and its possible correlation with ant attendance.

## 2. Material and Methods

### 2.1. Species Diversity and Sample Collection

We collected aphids during the seasons 2019–2021 in Poland, mainly from the Silesia region. The samples were collected into EtOH with a dilution of 70% for a slide preparation and to absolute EtOH 99.98% for molecular tests. Immediately after collection, we stored insect samples for NGS in absolute EtOH at −30 °C awaiting further analysis. From each sample containing aphids, we used up to three specimens for species identification, combining COI barcoding with morphological identification using Blackman & Eastop key [19]. Specimens are stored at the Entomological Collection of the University of Silesia in Katowice, Poland, are available upon request. The total number of 44 samples were collected, comprising 12 aphid species, unequally represented due to difficulties in finding particular species in field. Details of the collected samples are presented in Appendix A.

The data on myrmecophily of aphids existing in the literature are diverse and for particular species may differ, and the assessment of the degree of myrmecophily of particular aphid species may be misleading, depending on many factors [20]. While most *Dysaphis* species are regarded as being myrmecophilous, there is huge uncertainty on whether particular species are facultative or obligatory myrmecophiles, especially taking into account some life strategies (e.g., living in galls or underground). We analysed ant attendance only on the basis of field observations, noting whether the collected sample was attended to by ants or not, in other words, whether any ant worker was present within the aphid colony when uncovered. In case of aphids within curled leaves, the presence of ant workers inside the pseudogall was informative enough to treat it as ant attendance. In the case of leaf sheaths, either the presence of ants inside or soil cover made by ants was also indicative. In the case of underground living species, the presence of ant workers within the aphid colony after digging of the plant was also indicative. Such attitude definitely influenced the representativeness of samples in reference to ant attendance but also illustrated the general tendency of *Dysaphis* to myrmecophily.

### 2.2. DNA Extraction

Each DNA extraction was performed on a pool of two to five individuals (adults or 3rd-stage larvae) from the same colony rather than a single individual to reduce the risk of missing symbiotic infection. Prior to extraction, all specimens were surface-sterilised by individual immersion for 30 s in 5% bleach, followed by 30 s in 1 × PBS solution (NaCl 137 mmol/L). Insect DNA was extracted using an Sherlock AX extraction kit (A&A Biotechnology) abiding by the manufacturer’s protocol. The identification of bacterial associates of the examined species was conducted on the basis of the sequences of their 16S rRNA genes, the 16S rRNA gene was amplified using the following universal, eubacterial primers: 8F and 1541R [21,22], following the instructions of the manufacturer. For insect species identification, the primers LCO and HCO [23] were used to amplify the target 658 bp fragment of the cytochrome c oxidase subunit I (COI) gene. The PCR assays were performed in a final volume of 15 μL, containing 1 μL of genomic DNA, 0.5 μM of each primer, and a mixture for PCR containing Taq polymerase 2× concentrated (A&A Biotechnology). The thermocycling profile consisted of 94 °C for 1 min; 6 cycles of 94 °C for 1 min, 45 °C for 1 min and 30 s, and 72 °C for 1 min and 15 s; followed by 36 cycles of 94 °C for 1 min, 51 °C for 1 min and 30 s, and 72 °C for 1 min and 15 s; with a final 5 min extension period of 72 °C.

### 2.3. Next-Generation Sequencing and Sequence Processing

DNA concentration was measured prior to the library preparation procedure using the fluorimetric method using PicoGreen reagent (Life Technologies). The measurement was performed on the company’s Infinite apparatus Tecan. All samples passed the quantitative control positively. Specific sequences were used to amplify the selected region and prepare the library primers 341 F and 785 R [24]. The PCR reaction was performed using Q5 Hot Start High-Fidelity 2X Master Mix (New England Biolabs), with reaction conditions as recommended by the manufacturer. Sequencing was carried out on the MiSeq sequencer using the paired-end technology (ang: paired-end; PE), 2 × 300 nt, using the MiSeq Reagent Kit v3 (600-cycle) (Illumina), as per protocol.

Bioinformatic analysis comprised the highly variable region V3-V4 of gene 16S rRNA, providing classification of records to the species level, and was carried on with QIIME programme referring to the base of reference sequences Silva v138. Adapter sequences were removed with Cutadapt, which also served to analyse the quality of records and the removal of sequences of low quality (minimum length: 30). Sequences serving for OTU determination were 219–444 pb long. Paired sequences were joined with the algorithm fastq-join, and chimeric sequences were removed with the algorithm usearch61. Sequences were clustered with the algorithm uclust on the basis of reference base and taxonomic classification, including trees, conducted with algorithms uclast and fasttree. Additional analyses were performed with R software, suites phyloseq and vegan. Diagrams were generated using suites ggplot2, gplots, plotply, heatmaply, and metacoder. Classification of operational taxonomic unite and their numbers was presented in biom format. Alpha diversity analysis was performed with the application of the following factors: observed (a number of observed OTUs in a sample), Chao1, Shannon, and Simpson. Beta diversity analysis was performed with the application of the following metrics: Bray–Curtis, Jaccard, Weighted, and unweighted Unifrac, and for all samples, heatmap diagrams were clustered with the UPGMA method. As facultative symbionts, only bacteria with an abundance of records in single sample ≥1% were regarded. All data were placed in the Genbank database (submission SUB12083328) (list in Appendix A).

## 3. Results

Relative abundance of bacteria.

During the studies, a total number of 997 bacterial OTUs was recorded (Appendix A), classified into 239 bacterial species. The total number of recorded species per sample ranged from 8 to 122, with an average of 29.14 per sample. The mean number of recorded species did not significantly differ between ant-attended (28.14) and -unattended (31.54) samples but differed between samples from primary (19.45) or secondary (40.37) hosts (Figure 1, Appendix A). Only 10 genera constituted at least 1% of all recorded sequences per sample, with various abundances and presences in the studied samples (Table 1). *Buchnera aphidicola* was present in all studied samples, with shares ranging from 34.22% (*D. sorbi*, sample 12–21) to 99.99% (*D. plantaginea*, sample 70) and a mean share of 94.40%. It was the only symbiont (share of records ≥1%) in 27 of 44 studied samples.

Among the secondary symbionts present in the remaining 17 samples (Appendix A), the most common (in five samples: *D. brancoi*, *D. lappae*, two samples of *D. ranunculi*, and *D. lauberti*) was *Regiella*, with its share ranging from 4.78% to 27.68%, and *Fukatsuia* (in four samples, all *D. crataegi*), with its share from 10.73% to 14.78%. While the former was present only in samples from secondary host plants, the latter was found in samples from both primary and secondary host plants. Only in three samples was *Serratia* present: *D. plantaginea, D. brancoi*, and *D. sorbi,* in the latter (sample 12–21) reaching 65.41% of records. Moreover, *Spiroplasma* was present in two samples: *D. leefmansi* and *D. anthrisci*, at 1.02% and 7.70%, respectively. Other symbiotic bacteria occurred only in single samples.

Taking into account the frequency of records of secondary symbionts in particular species, the following results were obtained: *D. foeniculus* and *D. pyri*, single samples—0.00% of secondary symbionts; *D. plantaginea*, 2 of 14 samples—14.29%; *D. sorbi* 1 of 5—20.00%; *D. leefmansi* 1 of 4—25.00%; *D. brancoi* 1 of 3—33.33%; *D. crataegi* 5 of 8—62.50%; *D. ranunculi* 3 of 4—75.00%; and single samples of *D. anthrisci*, *D. lappae*, *D. lauberti*, and *D. newskyi* all comprised symbionts—100.00%.

There were only two samples with coinfections: *Serratia* (1.51%) with *Regiella* (4.78%) in ant-attended *D. brancoi* on *Valeriana officinalis* (sample 13d21) and *Regiella* (27.68%) with *Hamiltonella* (14.66%) and *Acinetobacter* (1.59%) unattended by ants *D. lappae* on *Arctium lappa* (sample 22–19). No coinfection of *Fukatsuia* and *Hamiltonella* was recorded.

### 3.1. Specificity of Aphid Symbiotic Microbiomes with an Indication of the Relationship with Ants

Taking into account shares of primary and secondary symbionts in recorded sequences, it is clear that aphid species tended by ants were less infected, regardless of the primary or secondary host plant (Figure 1 and Figure 2). *Buchnera* constituted 97% of recorded sequences in ant-attended samples while it was not attended by ants only in 88%. The shares of particular secondary symbionts also varied between ant-attended and not attended species (Figure 3). *Hamiltonella* was absent in ant-tended species, and the share of *Serratia* was very low, but *Wolbachia*, *Gilliamella*, *Sphingomonas,* and *Pelomonas* occurred only in species attended by ants.

Alpha diversity of symbionts (Table 2) was also affected by ant attendance. In the case of samples from the primary host, there were significantly lower values of alpha diversity indicators, and the mean number of observed sequences was in ant-attended colonies two times lower than in unattended colonies. A similar relation affected the values of other diversity indicators. The opposite situation applied to secondary hosts, wherein ant-attended colonies reached higher mean values of observed sequences and of the Chao1 indicator, but Simpson’s and Shannon’s values were almost identical.

Beta diversity revealed by heatmaps was most informative in the case of the analysis of OTUs with Bray and weighted Unifrac methods. In both cases, three main groups of samples were clustered apart from six or seven separate samples. In the case of the Bray method, there were three main clusters (Figure 4) and six samples of significant separateness. In case of the weighted Unifrac method, we were able to distinguish five clusters and seven separate samples (Figure 5). However, none of these clusters correlated with myrmecophily, comprising samples of both attended and unattended aphid species. Rather, in the case of the Bray method, existing clusters corelated with the seven most often recorded *Buchnera* OTUs (Figure 6), while in the case of the weighted Unifrac method (Figure 6), at least two of the distinguished clusters—C and D—were cluster samples comprising *Fukatsuia* and *Regiella*, respectively (Figure 7).

### 3.2. Specificity of Aphid Symbiotic Microbiomes in Reference to Life Cycle and Host Plant Affiliation

Taking into account shares of primary and secondary symbionts in recorded sequences, it is clear that aphid species on primary hosts were less infected than on secondary hosts (Figure 1 and Figure 2), although the difference was not as high as in the cases lacking ants or with the presence of ants. On primary hosts, *Buchnera* constituted 95.48% of the recorded sequences, while on secondary hosts, it was 93.47%. Furthermore, only three secondary symbionts occurred on primary hosts: *Serratia* (two samples, mean share of 2.96%), *Fukatsuia* (2, 1.16%), and *Gilliamella* (1, 0.13%), whereas a total number of nine secondary symbionts occurred on secondary hosts: *Serratia* (1, 0.07%), *Regiella* (5, 3.21%), *Fukatsuia* (2, 1.17%), *Hamiltonella* (1, 0.70%), *Wolbachia* (1, 0.25%), *Spiroplasma* (2, 0.42%), *Sphingomonas* (1, 0.06%), *Pelomonas* (1, 0.09%), and *Acinetobacter* (1, 0.08%) (Figure 8). Of these, only *Serratia* and *Fukatsuia* occurred on both primary and secondary hosts, while *Gilliamella* occurred only on primary host of *D. plantaginea*. Also noticeable was the presence of *Fukatsuia* solely in *D. crataegi*, regardless the host plant, although not in all samples.

The share of symbionts in monoecious and heteroecious species was relatively similar, constituting 6.72% and 5.16% of recorded sequences in 9 and 35 samples, respectively. Among them, in monoecious species, *Regiella* (3.61%) and *Hamiltonella* (1.63%) dominated, while in heteroecious species, *Serratia* (92.96%) and *Fukatsuia* (1.46%) dominated. This was related to the aphid species biology because *Regiella* was very abundant in monoecious *D. lappae* (27.68%), although it was also present in heteroecious *D. ranunculi* and *D. lauberti*. In contrast, *Serratia* was predominant in heteroecious *D. sorbi* (65.41%) and *Fukatsuia* also in heteroecious *D. crataegi* (four of eight samples, 6.40%).

Alpha diversity of symbionts was very differentiated. On primary hosts belonging to Rosaceae, except of *Crataegus,* the diversity indicators reached relatively low values (Table 3). In *D. plantaginea*, all the samples from *Malus* as well as the single sample from *Heracleum* had similarly low values of observed OTUs. In *Crataegus*, the number of observed OTUs in *D. crataegi* was up to six times higher than in the remaining primary hosts, and also it was the highest among all species collected from secondary hosts (Table 4). Samples of other aphid species from secondary hosts also reached much higher values of observed OTUs than any species on the primary host, except for *Crataegus*.

The beta diversity revealed by heatmaps was most informative in the case of the analysis of OTUs with the Bray method, where the clusters A and B, together with single samples (10–19, 12–21, 36–19, 13–21), comprised only aphids collected from the primary host (Figure 9), except of sample 8d-21 collected from the secondary host. The remaining clusters consisted of samples from either the primary host (few) or secondary host (predominant).

## 4. Discussion

The results show great diversity and variability of facultative bacterial symbionts within the aphid genus *Dysaphis*. The relationship with the host plant via life cycle is a leading and the strongest factor influencing the presence and composition of secondary endosymbionts in the studied species; however, ant attendence may significantly influence this relationship. First of all, it seems that aphids on primary hosts mainly rely on symbiosis with *Buchnera aphidicola*, being additionally supported by *Serratia* or *Fukatsuia*. Regarding primary hosts as ancestral or at least an evolutionary older nutritional source for aphids, relaying on *Buchnera* is not surprising, as it has an important contribution in the exploitation of nutritional components of phloem sap. The additional presence of *Serratia* and *Fukatsuia* may enhance resistance to parasitoids, because on primary hosts, aphids live in more open spaces (even in pseudogalls) and are exposed to more variable atmospheric conditions, which may temporarily decrease the protective presence of ants. However, the presence of both secondary endosymbionts was confirmed in both ant-attended and unattended samples. The results agree with data from Henry et al. [17] in terms of the higher presence of *Hamiltonella* in unattended species (Figure 3), and the same pattern also affects *Regiella* and *Fukatsuia*. The opposite pattern concerned *Serratia*, which also reached very high abundance in an unattended colony of *D. sorbi* on the primary host but was also present in the attended colonies of *D. brancoi* and *D. plantaginea*. Yet, the results were opposite to those of Mandrioli et al. [16] who recorded *H. defensa* and *R. insecticola* in 37.5% and 12% of studied specimens of ant-attended *D. plantaginea* on *Malus* sp., respectively. In our studies, these bacteria were absent in all 14 studied samples of this aphid species, which indicates very high variability of infections by these symbionts. Moreover, general results by McLean et al. [13] differed, indicating *Hamiltonella*, *Serratia,* and *Regiella* in 50%, 35%, and 22% of a total 70 studied species respectively, while among 12 *Dysaphis* species, the results showed *Regiella*, *Serratia,* and *Spiroplasma* in 33%, 25%, and 17% of species, respectively. *Dysaphis* comprises mostly heteroecious species, changing host plant species during the life cycle, sometimes utilising various secondary hosts, e.g., *D. crataegi* [19], which may contribute to the observed variability of bacterial infections. Such differences were already recorded in *Schlechtendalia chinensis*, with the host alternating being between mosses and sumac [25]. It is possible also that some environmental factors, e.g., climate or the presence of other insects such as honeybees (*Apis mellifera*), may contribute to this variability. Bees are the dominant and most abundant pollinators of apple trees and are proven to possess *Gilliamella* as an element of their gut microbiota [26]. Possible contamination of aphid colonies by excretes of bees or by ant workers having previous contact with bees could have contributed to the infection of aphids by *Gilliamella*.

It seems that strains of *Hamiltonella defensa* and *Regiella insecticola* may variously influence the composition of aphids’ cuticular hydrocarbons [27], which is further informative for mutualistic ants. It is proven that aphids recognise the difference of symbiont-related odour of aphids, and on this basis, they may discriminate efficiency of mutualistic aphids in relation to endosymbiont strains. In our case, both these endosymbionts were present in aphids feeding on secondary hosts, irrespective of ant attendance. The sample of *D. lappae* comprised both these endosymbionts (the only sample with *Hamiltonella*), but it was not attended by ants, while all other aphid species infected with *Regiella* applied to ant attended colonies. *Regiella* was the most common secondary endosymbiont present in colonies feeding below the soil level or in ant chambers on three different hosts plants (*Valeriana*, *Pastinaca*, *Ranunculus*). Such relation of *Regiella* to feeding location of aphids may support the influence of this symbiont on the protection of aphids against fungal infections [7,28,29]. While living in cryptic niches under soil and in the constant presence of ants may be sufficient protection against parasitoids, there still may be significant exposure to fungal infections in such aphids.

The results confirm the protective influence of ants against pathogens and parasitoids of aphids and their excluding influence on infection by secondary endosymbionts. This could explain the results by Ivens et al. [18] on *Prociphilus* aphids, where within nine ant-tended species, only one possessed *Serratia* as a single facultative symbiont recorded in myrmecophilous aphids of that study. However, there are suspected cases of transfer of bacterial symbionts from aphids to ants, and further by ants to the environment, and then through plants again to aphids [30,31], also suspected by Ivens et al. [18]. Generally, it seems that environmental conditions serve as a reservoir of potential facultative endosymbionts, e.g., *Serratia symbiotica* [30], with constant multidirectional flow of these bacteria between various hosts and their acquisition through contact between these hosts, e.g., aphids may be infected by parasitoids [32]. If the latter case was relatively common, then again ants might protect aphids from infection if aphid species are obligatorily myrmecophilous.

Nevertheless, the ant-attended colonies of aphids on secondary hosts were characterised by the highest values of alpha diversity (Table 2) with the presence of seven bacterial genera, while on primary hosts, there were only two, with *Serratia* common in both cases (Figure 2). This result may put into question whether ants inhibit bacterial infection because unattended colonies on secondary hosts comprised only four bacterial genera, with *Regiella* and *Hamiltonella* being the most abundant (Figure 2). While in unattended colonies, the most abundant were the most common aphid symbionts: *Serratia*, *Regiella*, *Hamiltonella*, and *Fukatsuia*, in ant-attended colonies, there were more bacterial genera but they occurred with lower abundance: *Wolbachia*, *Spiroplasma*, *Sphingomonas*, and *Pelomonas*. It is difficult to conclude whether these were just occasional, random infections or whether the environmental conditions fixed by ants facilitated infections by these bacteria. None of them are new, and they were reported to infect aphids in other studies [9,10,13,17], but their role and influence on aphid ecology is poorly known. It must be highlighted that feeding locations of *Dysaphis* on secondary hosts are placed in leaf sheaths under soil cover made by ants or directly on roots in ant chambers. Such proximity of soil may somehow enhance bacterial infections, despite the presence of ants, and while mutualism may inhibit symbiosis with common bacterial symbionts, it may not fully be secure from random infections by other bacteria present in the environment.

It is also difficult to unambiguously correlate the beta diversity of acquired facultative symbionts with ant–aphid mutualism or aphid species phylogeny, although samples of *D. plantaginea* were clustered together, irrespective of host plant. Other species were clustered variously, depending on the method, and clusters did not comprise all representatives of the species. None of the secondary symbionts was restricted to a single aphid species. Such differentiation of symbionts seems to indicate multiple, independent acquiring of facultative bacteria. Deviation from this general view was *D. crataegi*, which possessed *Fukatsuia* in four of eight samples, both from primary and secondary hosts. *Fukatsuia* and *Regiella* are suspected to provide heat stress tolerance for *Buchnera*, which has significant importance for aphid survival [33]. While the former occurred on both primary and secondary hosts, the latter was found only on secondary hosts. If *Regiella* is also responsible for protection against fungi (and at least one strain against parasitoids [34]), then its occurrence in the proximity of soil may be related to its function. In the case of *Fukatsuia*, such a correlation of function and aphid location cannot be traced. Similarly, no correlation with ant–aphid mutualism could be observed in both symbionts.

The significant differences in symbiont infections in studied species of *Dysaphis* showed huge variability of this type of mutualism. We found no correlation between the composition of symbiont species and host plant species, neither with aphid species nor with ant–aphid mutualism. The clear difference was only between the abundance and diversity of symbionts on primary and secondary hosts. The studies conducted thus far indicate environmental versality of facultative bacterial symbionts and various routes of infections of insects, also via host plants. The presented results strongly corroborate these findings, and if life cycle strongly contributes to symbiosis with bacteria, then the obtained results cannot be surprising, taking into account heteroeciousness of *Dysaphis* and utilisation of various plants as secondary hosts. It seems that exploitation of the primary host is more influential and decreases the ratio of infection by facultative symbionts. By contrast, utilisation of the secondary hosts allows more diverse infections. The presence of ants inhibits the infection quantitatively, proving their protective role and, to some extent, their interchangeable character with symbionts in protection of aphids. Simultaneously, the microhabitat made by ants somehow increases the diversity of observed bacterial infections, and possibly ants intermediate in migration of symbionts from aphids to the environment. The variability of symbionts in heteroecious aphids requires further investigations, because if the life cycle of aphids (through various modes) drives infections, then host alternation and change of habitat may strongly affect the acquisition of new bacterial symbionts by such aphids.

## Figures and Tables

**Figure 1 insects-13-01089-f001:**
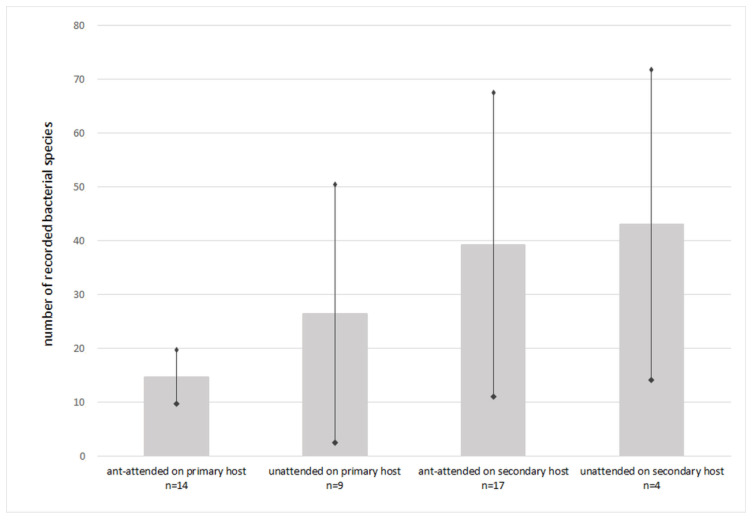
Mean number of bacterial species recorded in particular groups of samples (vertical bars indicate ± SD).

**Figure 2 insects-13-01089-f002:**
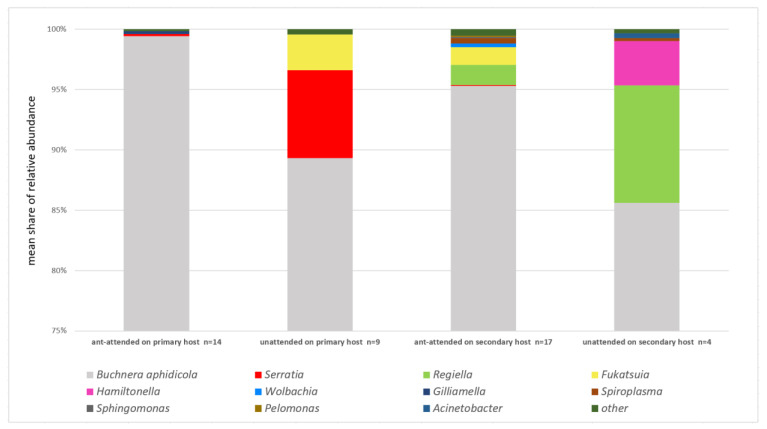
Shares of endosymbionts on aphid hosts in reference to ant attendance.

**Figure 3 insects-13-01089-f003:**
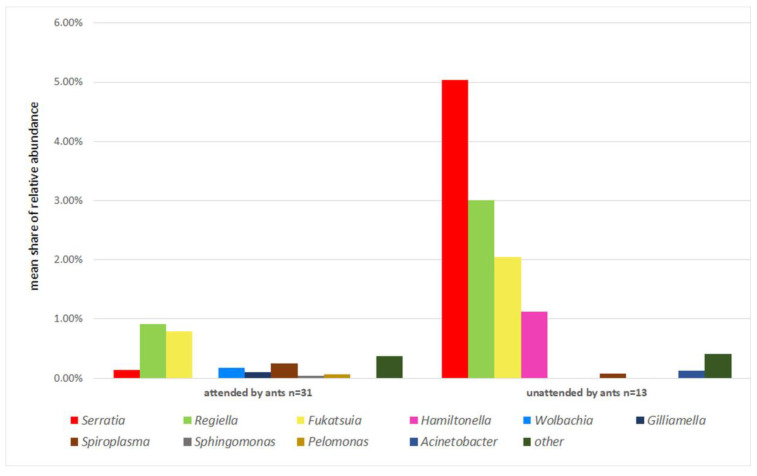
Mean shares of particular secondary symbionts in reference to ant attendance.

**Figure 4 insects-13-01089-f004:**
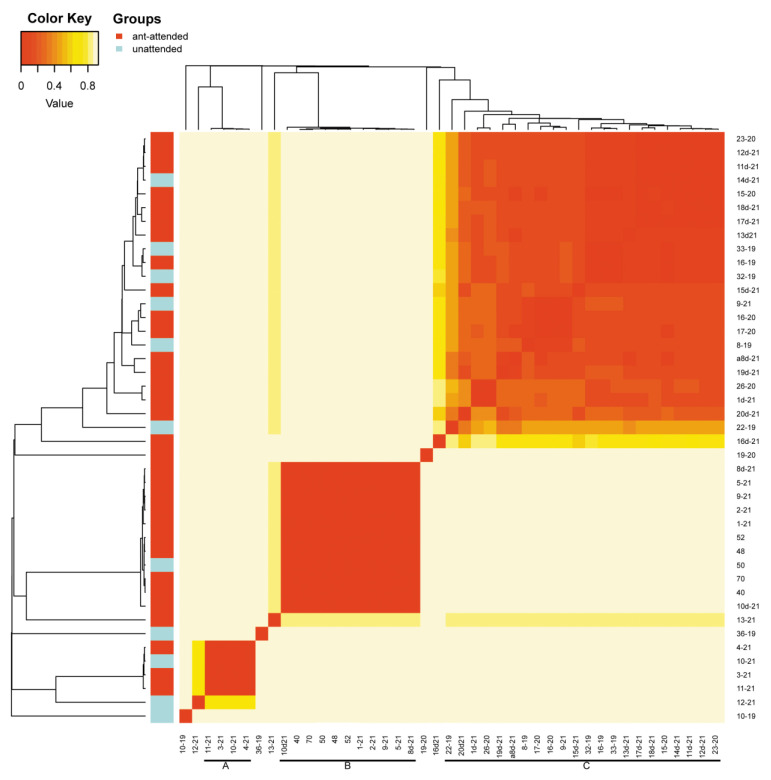
Heatmap of OTU beta diversity clustered with the Bray method in reference to the ant attendance of studied samples (sample numbers as in Appendix A; bars with letters mark clades).

**Figure 5 insects-13-01089-f005:**
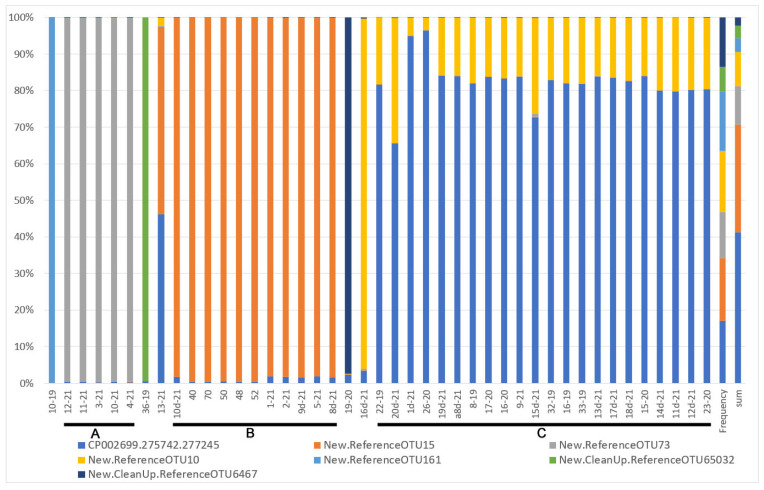
Shares of the 7 most abundant OTUs of *Buchnera aphidicola* in the studied samples. The order of samples follows the order of samples on the x-axis of the heatmap shown in Figure 4 (bars and letters indicate the same clades as those in Figure 4).

**Figure 6 insects-13-01089-f006:**
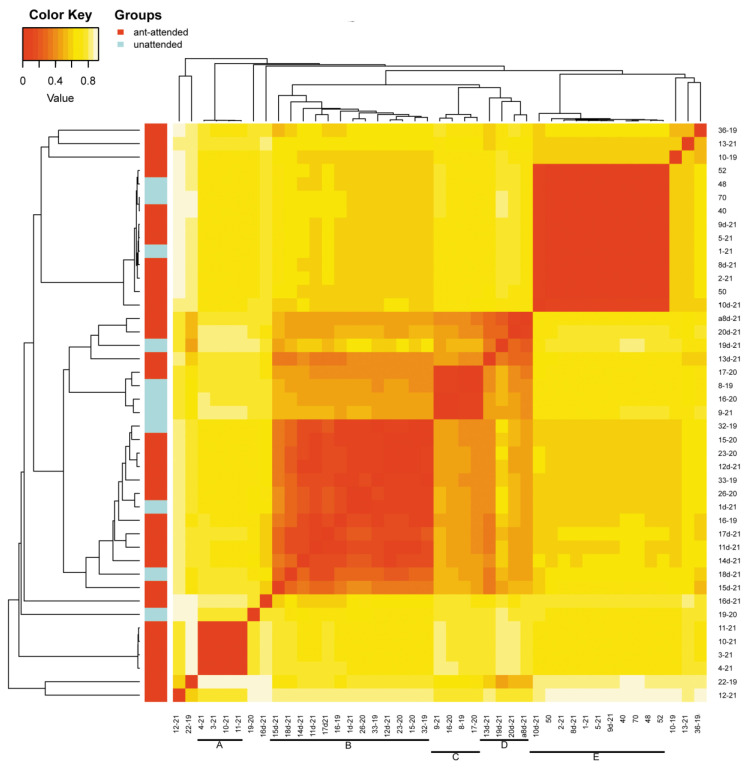
Heatmap of OTU beta diversity clustered with the weighted Unifrac method in reference to ant attendance of the studied samples (sample numbers as in Appendix A; bars with letters mark clades).

**Figure 7 insects-13-01089-f007:**
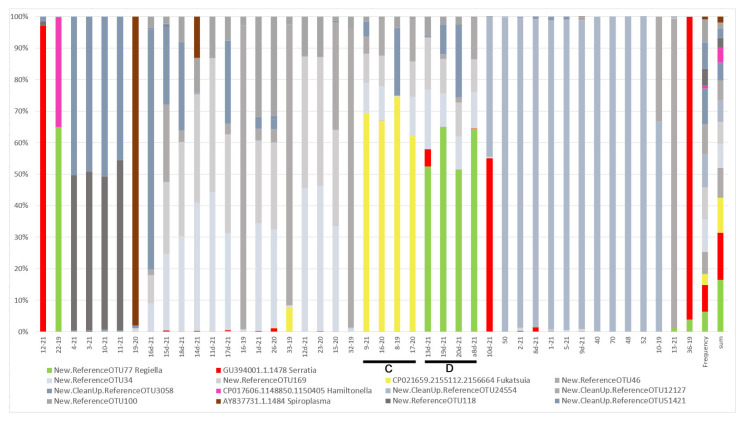
Shares of the next 14 most abundant OTUs with abundance >0.1%—*Buchnera* and secondary symbionts (affiliation with the genus presented at the name of the OTU, with the unnamed ones belonging to *Buchnera*) in the studied samples. The order of samples follows the order of samples on the *x*-axis of the heatmap in Figure 6 (bars and letters indicate the same clades as in Figure 6).

**Figure 8 insects-13-01089-f008:**
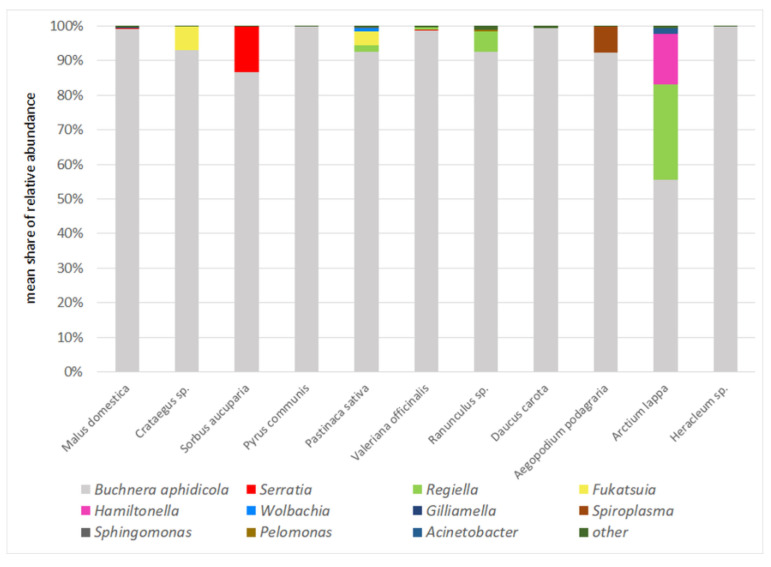
Mean shares of particular endosymbionts in reference to host plants (*Malus*, *Crataegus*, *Sorbus,* and *Pyrus* are primary hosts, the remaining plants species are secondary hosts).

**Figure 9 insects-13-01089-f009:**
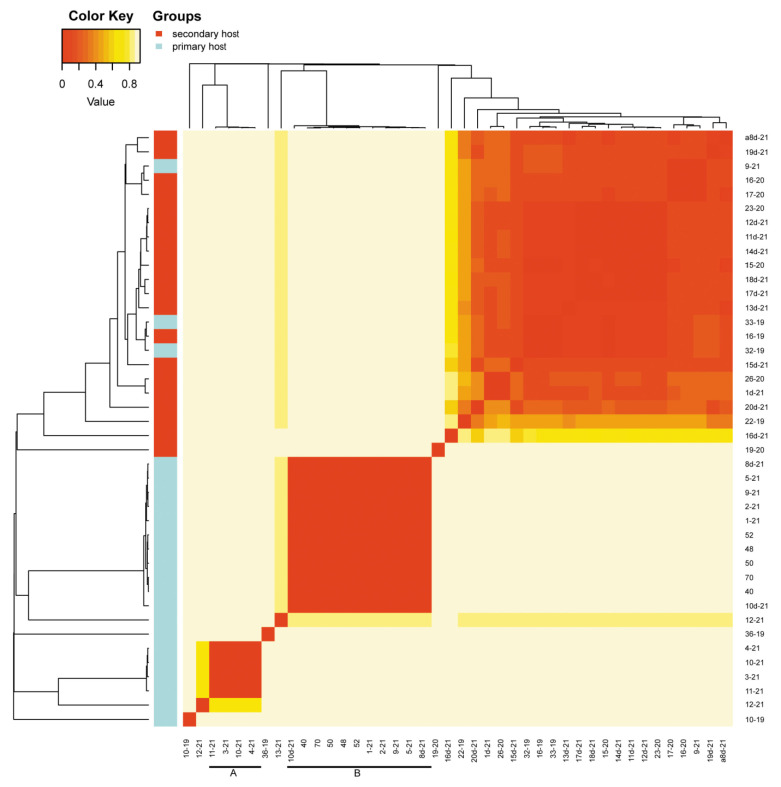
Heatmap of OTU beta diversity clustered with the Bray method in reference to host plants of studied samples (sample numbers as in Appendix A; bars with letters mark clades).

**Table 1 insects-13-01089-t001:** Records of symbiont species in the studied samples.

OTU ID	Total No. of Records	Mean No. of Records	Relative Abundance (%)	Presence in No. of Samples
*Buchnera aphidicola*	4,012,657	93,318	94.40	44
*Regiella*	67,882	1579	1.59	21
*Serratia*	62,685	1458	1.60	28
*Fukatsuia*	50,653	1178	1.24	31
*Hamiltonella*	17,991	418	0.34	2
*Spiroplasma*	8211	191	0.20	3
*Pelomonas*	2990	70	0.08	25
*Acinetobacter*	2042	47	0.04	16
*Wolbachia*	1902	44	0.12	3
*Gilliamella*	1464	34	0.07	1
*Sphingomonas*	277	6	0.01	11

**Table 2 insects-13-01089-t002:** Values of alpha diversity indicators of symbionts in reference to ant attendance and host plant.

	Observed	Shannon	Simpson	Chao1
	Primary Host			
Attended	80.21	0.34	0.12	106.37
Unattended	162	0.68	0.28	190.08
	Secondary Host			
Attended	240.71	1.11	0.45	304.40
Unattended	196.75	1.06	0.45	240.77

**Table 3 insects-13-01089-t003:** Values of alpha diversity indicators of symbionts in reference to host plant species.

	Observed	Shannon	Simpson	Chao1
	Primary Hosts			
*Malus domestica*	88.38	0.33	0.12	117.09
*Crataegus sp.*	297.67	1.07	0.44	348.35
*Sorbus aucuparia*	65.60	0.52	0.22	77.93
*Pyrus communis*	49.00	0.16	0.05	53.00
	Secondary Hosts			
*Valeriana officinalis*	151.29	0.77	0.33	215.12
*Ranunculus sp.*	307.50	1.35	0.53	368.99
*Pastinaca sativa*	288.71	1.28	0.51	344.55
*Arctium lappa*	206.00	1.64	0.72	236.00
*Aegopodium podagraria*	110.00	0.54	0.21	228.46
*Heracleum sp.*	93.00	0.27	0.09	135.00
*Daucus carota*	316.00	0.93	0.39	330.51

**Table 4 insects-13-01089-t004:** The values of observed OTUs in particular species in reference to host plants (*Malus*, *Crataegus*, *Sorbus,* and *Pyrus* are primary hosts, and the remaining plants species are secondary hosts).

Aphid Species	*Malus domestica*	*Crataegus sp.*	*Sorbus aucuparia*	*Pyrus communis*	*Valeriana officinalis*	*Ranunculus sp.*	*Pastinaca sativa*	*Arctium lappa*	*Aegopodium podagraria*	*Heracleum sp.*	*Daucus carota*	Mean
*D. brancoi*					188.00							188.00
*D. crataegi*		297.67					325.5					307.00
*D. foeniculus*											316.00	316.00
*D. lappae*								206.00				206.00
*D. leefmansi*					123.75							123.75
*D. plantaginea*	88.38									93.00		93.00
*D. pyri*				49.00								49.00
*D. ranunculi*						307.5						307.50
*D. sorbi*			65.60									65.60
*D. anthrisci*									110.00			110.00
*D. newskyi*							187.00					187.00
*D. lauberti*							262.00					262.00
mean	88.38	297.67	65.60	49.00	151.29	307.5	288.71	206.00	110.00	93.00	316.00	

## Data Availability

The NGS sequencing data used during the current study are available from the corresponding author on reasonable request.

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
