# Peer review of "Microbiome of the Aphid Genus Dysaphis Börner (Hemiptera: Aphidinae) and Its Relation to Ant Attendance"

_insects, 2022, doi:10.3390/insects13121089_

Round 1

Reviewer 1 Report

This paper deals with the relationships between the several life modes of aphids and the composition of the primary and secondary endosymbionts. In particular, the authors focused on the effects of the difference between generations on the primary hosts and the secondary host and the difference between the presence and absence of attending ants on the composition of symbionts. This is an interesting attempt and the authors obtained obvious results. I think that this paper is potentially worth publishing. However, this paper has several problems at present; sentences are lengthy and confused, including several grammatical problems and misspelling. It is needed to improve the quality of English itself. I recommend the authors to condense the parts of Introduction and Discussion approximately to the half of the present volume. English-editing by any native English speaker may be needed.

The following is minor comments,

Line 29. Suggestion; Cite Shigenobu et al. 2000 (Nature volume 407: 81–86 2000) in addition to Chong.

Line 30. Suggestion; Cite Clark et al. 2000 (Evolution 54: 517–525) in addition to Novakova. 

Line 46. “the acquisition of” may be better than “aquainting”.

Line 48. “phylogeny” may be better than “relatedness”.

Line 55. “apart from….parasitoids,” may not be necessary.

Line 60. “myrmecophilic aphids” may be clearer than “myrmecophiles”.

Line 63-66. It is difficult to understand. Reword the sentence to increase clarity.

Line 71-74. It is difficult to understand. Reword the sentence to increase clarity.

Line 81-85. It is difficult to understand. Reword the sentence to increase clarity.

Line 119. Remove the period.

Line 220. “corelated” may be “correlated”.

Line 222. Fig.5 may be Fig.6.

Line 223. Fig.6 may be Fig.5.

Figure 6. I recommend the authors to indicate the positions of clusters A, B, and C in the figure.

Figure 7. I recommend the authors to indicate the positions of clusters C and D in the figure.

Line 255. For 6.72% and 5.16%, what is the denominator.  

Line 257. For “Serratia 92.96%)”, “Serratia (92.96%)” may be correct.

Line 259. “while” should be “in contrast,”

Line 265. both of samples from Malus and a single sample from…? 

Table 4. Clarify which plant genus is the primary host or secondary host in the table.

Line 277. Clarify which is cluster A or B in Fig. 9.

Discussion is lengthy and confused, and difficult to understand, including grammatical problems. Reword the sentence to increase clarity. Consider using different words rather than “concern”.

Line 302. “concerns” may be “is found in”.

Line 303. “follow” may be “agree with”.

Line 320. “isnects” may be “insects”.

Line 327. Aphids? This should be ants.

Line 330. “not necessarily…” may be “irrespective of the presence or absence of…”

Line 363. “be” may be “by”.

Line 390-391 and 396-400. The two sentences seem to be contradictory with each other. The readers will be confused.

---

Author Response

We would like to thank the reviewers for all comments and suggestions, which greatly influenced the quality of the manuscript. We followed the concerns carefully and corrected all the recorded errors and mistakes.

We attempted to shorten the Introduction and Discussion, although it was difficult to omit the speculative passages of the Discussion. The results of our study support great variability of aphid microbiome and we tried to find possible, often hypothesised solutions but we believe it is proper for Discussion section to indicate possible explanations just to show further areas for research. 

We also revised the text linguistically. 

Reviewer 2 Report

The article "Microbiome of the aphid genus Dysaphis Börner (Hemiptera: Aphidinae) and its relation to ant attendance" is devoted to the study of the infection of species of this aphid genus with symbiotic bacteria. The correlations of the detection of facultative symbionts in Dysaphis aphid with the plant species and the relationship with ants are considered.

A total of 44 pools of aphid specimens were examined. Identification of 12 aphid species was done based on COI sequences. Microbiome of 44 samples was studied using Illumina HiSeq 4000 with V3-4 region  of 16S rRNA.

Mean number of recorded OTUs did not significantly differ between ant attended (28.14) and unattended (31.54) samples but differed between samples from primary (19.45) or secondary (40.37) hosts. Nine species of facultative symbionts were found in ant-attended, and 7 species in unattended ones. Only 3 recognized species of secondary symbionts were found in aphid specimens from primary plants and 9 bacterial species in aphid specimens from secondary plants.

The such study design is particularly interesting and could bring important findings about co-evolutionary relations between aphids, symbiotic bacteria and ants. I have several comments, which should be considered or explained by the Authors in the revision of their work:

SM2 and SM3 missing in Suppl. mat.

Material and Methods

According to the given data, it is not clear how long the aphid colonies were observed on the attendance of ants. Without this, the classification regarding the association with ants is doubtful.

 Reference needed for primers: "eubacterial primers: 8F and 1541R" and also "the library primers 341F and 785R". These primers have authors.

L137 "highly variable region V3-4 of gene 16S rRNA, providing classification of records to the species level, and was carried on with QIIME  programme referring to the base of reference sequences Silva v. Adapter sequences  were removed with Cutadapt, which also served to analyse quality of records and removal of sequences of low quality (minimum length 30)." But, usually, the analyzed length is 200-300 bp. It is unacceptable, if authors worked with 30 bp seqs.

Results

It is very difficult to comprehend the numbers of infected samples in the text, figures and tables. Everything is confusing. There is a lack of one table that would combine data on aphid species, their symbionts and relationships with ants.

 L193 Sphingomonas, but in Table 1 is Spiroplasma

In Fig. 9 there is no designation of clusters A and B.

 Different aphid species are infected with different symbionts. And if the studied aphid species is infected, then the infection is present on different plants, i.e. no regular influence of plant species on the number of symbionts was found. For example, those aphid species, D. crataegi and D. plantaginea, which were studied from different types of plants, are equally infected with symbionts both on the primary and secondary plants (Tables 4, 5).

 If all 44 samples are taken into account (regardless of the plant type), 9 species of facultative symbionts were found in ant-attended, and 7 species in unattended ones (Fig. 3). This cannot be considered a significant difference.

 The total number of "44 samples were collected, comprising 12 aphid species, unequally represented". Since in some cases 1 sample per aphid species was studied, this amount of graphical information is completely redundant. Species were clustered variously depending on the used method. And the lack of information on the length of time ants and aphids have been observed makes it difficult to unambiguously correlate the diversity of acquired facultative symbionts with ant-aphid mutualism.

The main result of the work is that the share of facultative symbionts was related to the aphid biology.

 Discussion - general comment

I think that some parts of the discussion are too speculative.

The authors state that "The results clearly indicate that relationship with host-plant is a leading and strongest factor influencing the presence and composition of secondary endosymbionts in the studied species of Dysaphis" (L291). But their results prove that infestation with symbionts (in cases where one aphid species was studied from different types of plants) does not depend on the plant species.

None of the secondary symbionts was restricted to a single aphid species. The results of this work and previously published data indicate very high variability of infections by symbionts in Dysaphis aphids, which is justified by the biology of Dysaphis aphids. After all, authors stated: it was found no correlation between symbionts and host plants, neither with aphid species nor with ant-aphid mutualism. This contradicts their own statements.

However, questions raised by the authors about symbionts variability in heteroecious aphids and ant-aphid mutualism require further investigations.

Minor:

There are many mistakes, the text needs to be corrected

attndence

Maus domesticus

tchem

Author Response

(The authors gave the same response as above.)

Reviewer 3 Report

Aphids establish very interesting mutualistic symbiosis with ants in natural ecosystems. This paper aims to study the effects of ants on aphid endosymbiont species and diversity, this idea is very novel and interesting. However, there are some problems in writing and data analysis. In particular, the relevant information of the samples is not clear, resulting in readers' inability to understand the analysis results of the data. The figures and tables lack explanation, and the information is not accurately expressed. Information about primary and secondary hosts is not explained in the method section, figures, or tables of the paper. The Supplementary-Material-one is not well made, and it is difficult to understand.

 Other comments:

Line 11, change “attndence” into “attendence”

Line 13, change “V3-4” into “V3-V4”, similar correction in whole manuscript.

Line 96-97, It is difficult to find 12 aphid species in the Suplementray-Material-one. So table of Suplementray-Material-one should clearly show the sample’s information.

Line 113, “tchem”?  Please provide the concentration of PBS solution.

Line 119, change “.” into “,”

Line 155, Where is “SM3-out table L7”?

Line 161-163, The information mentioned in this sentence cannot obtain from Table 1, such as “D. sorbi, sample 12-21”, “D plantaginea, sample 70”, 94.52%

Line 166-184, which table or Figure can show this information?

In Figure 1, Figure 2, Figure 3, there are no figure legends.  Latin name of bacteria genera should be italic. The ordinate is not marked.

In Figure 4 and Figure 5 are difficult to understand, because the code of samples cannot find in sample information.

Author Response

(The authors gave the same response as above.)

Round 2

Reviewer 2 Report

The authors corrected the text and responded to the comments. However, unclear sentences remained in Discussion:

It is proven that aphids recognise 358 the difference and on this base may discriminate efficiency of mutualistic aphids in rela-359 tion to endosymbiont strains.

Sample of D. lappae comprised both these endosymbionts (the only one of Hamil-362 tonella) but it was not attended by ants while all other infections with Regiella concerned 363 applied to ant attended colonies.

The wording lacks an indication of aphids: The results confirm protective influence of ants against pathogens and parasi-372 toids and their excluding influence on infection by secondary endosymbionts.

L395 None of them is new and  were reported to infect aphids in other studies (McLean et al. 2019, Ma et al. 2021, He et 396 al. 2021) .. Authors cite only works of recent years. However, there are earlier publications that also summarize information about the discovered aphid symbionts, for ex. DOI 10.1016/j.aspen.2017.03.025 

Author Response

Dear Reviewer

thank you again for comments and suggestions to our manuscript. We did following changes:

It is proven that aphids recognise 358 the difference and on this base may discriminate efficiency of mutualistic aphids in rela-359 tion to endosymbiont strains.

clarified

Sample of D. lappae comprised both these endosymbionts (the only one of Hamil-362 tonella) but it was not attended by ants while all other infections with Regiella concerned 363 applied to ant attended colonies.

clarified

The wording lacks an indication of aphids: The results confirm protective influence of ants against pathogens and parasi-372 toids and their excluding influence on infection by secondary endosymbionts.

corrected

L395 None of them is new and  were reported to infect aphids in other studies (McLean et al. 2019, Ma et al. 2021, He et 396 al. 2021) .. Authors cite only works of recent years. However, there are earlier publications that also summarize information about the discovered aphid symbionts, for ex. DOI 10.1016/j.aspen.2017.03.025 

implemented

Reviewer 3 Report

The author has made more detailed revisions according to the first review comments, but there are still some problems with the details of the chart in the paper.

 Line 178, in Table 1, change “OUT ID” into “symbiont species”; relative abundance is mean? relative abundance of Buchnera aphidicola is 94.40%, but it is 94.52% in line 176.

Line 181, what is “2 x D. ranunculi”?

Line 179-198, where do these descriptions found? which table or figure?

Line 218, in Figure1, I don’t understand the number in y-axis, are they “0, 1000, 2000, … 8000” or “0, 10, 20, …, 80”?  if they mean “number of recorded bacterial species”, and it is contrary with the result (239 bacterial species) in line 169.

Line 225, in Figure3, the data in y-axis, Comma should be changed to dot. E.g. “0,00” change into “0”, “1,00%” change into “1.00%”, and so on.

Line 230, in Table 2, what does “observed” mean? Similar problem in Table 3.

Line 285, The presentation and form of Table 3 needs improvement, same with Table 2.

For Figure 6 and Figure 7, I don't see their practical significance to the paper.

Author Response

Dear Reviewer

Thank you for all suggestions and comments to our manuscript. Our response is following:

Line 178, in Table 1, change “OUT ID” into “symbiont species”; relative abundance is mean? relative abundance of Buchnera aphidicola is 94.40%, but it is 94.52% in line 176.

The correct value has been given into the text

Line 181, what is “2 x D. ranunculi”?

corrected

Line 179-198, where do these descriptions found? which table or figure?

Implemented – data available in SM1 table

Line 218, in Figure1, I don’t understand the number in y-axis, are they “0, 1000, 2000, … 8000” or “0, 10, 20, …, 80”?  if they mean “number of recorded bacterial species”, and it is contrary with the result (239 bacterial species) in line 169.

The y-axis presents the number of recorded species, which is mean value for each presented set of samples. The bars represent standard deviation. It is not contrary to line 169 because these are not qualitative data, and the species may not necessarily repeat in each sample. That is why the mean values may be lower than the total number of species.

Line 225, in Figure3, the data in y-axis, Comma should be changed to dot. E.g. “0,00” change into “0”, “1,00%” change into “1.00%”, and so on.

Corrected

Line 230, in Table 2, what does “observed” mean? Similar problem in Table 3.

Observed is a simplest index of measuring alpha-diversity, it is a number (in this case a mean number) of observed OTUs in a given set of samples. Explanation has been added to the M&M section.

Line 285, The presentation and form of Table 3 needs improvement, same with Table 2.

In our opinion the Tables 2 and 3 are as simple, clear and comprehensible as only it is possible.

For Figure 6 and Figure 7, I don't see their practical significance to the paper.

As stated in the Discussion, the figures show no correlation of existing beta-diversity neither with particular host plants as well as with ant-attendance and indicate also minute correlation with facultative symbionts. This shows that the factors influencing infections with endosymbionts are more complicated and unresolved in host-alternating species of aphids and as such – contribute to general conclusions of the MS.